# Proof-of-Concept Studies Demonstrate That Food and Pheromone Stimuli Can Be Used to Attract Invasive Carp So Their Presence Can Be Readily Measured Using Environmental DNA

Ratna Ghosal [1,2,*] , Alison A. Coulter [3] and Peter W. Sorensen [1]

1 Department of Fisheries, Wildlife and Conservation Biology, University of Minnesota, Twin Cities, MN 55108, USA; soren003@umn.edu
2 Biological and Life Sciences, School of Arts and Sciences, Ahmedabad University, Ahmedabad 380009, Gujarat, India
3 Department of Natural Resource Management, South Dakota State University, Brookings, SD 57006, USA; alison.coulter@sdstate.edu
* Correspondence: ratna.ghosal@ahduni.edu.in

**Abstract:** The utility of environmental DNA (eDNA) as a detection tool for fisheries management is limited by dilution and degradation, especially in areas of low fish abundance. This proof-of-concept study addressed these challenges by testing whether food or pheromones might be used to attract invasive carp so they can be measured more readily using eDNA. In two experiments, PIT-tagged carp were stocked into ponds (N = 3 for silver carp; N = 1 for common carp) while one of two stimuli (planktonic food [spirulina] for silver carp and a sex pheromone [prostaglandin F2$\alpha$] for male common carp) was added to determine if we could attract fish to one side while measuring both fish presence (detections) and eDNA concentrations. The addition of spirulina increased detections of silver carp by 2–3 fold, while eDNA concentrations increased by 4-fold on the test side when compared to the side without the stimulus. The addition of the sex pheromone increased detections of common carp by ~25-fold, where a 6-fold increase in eDNA concentrations was measured ($p < 0.05$). A strong positive correlation was noted between fish presence and eDNA concentration for both species. These experiments demonstrate that food and pheromone stimuli could be used to attract invasive carp so they could be measured more easily and accurately.

**Keywords:** silver carp; common carp; Aquatic Invasive Species (AIS); spirulina; eDNA; prostaglandin F2$\alpha$



## 1. Introduction

Surveys incorporating measurements of environmental DNA (eDNA) have improved managers' ability to detect invasive fish species [1–5]. However, eDNA detection rates can be limited by many factors, including dilution, fish movement, degradation, and the presence of dead fish/feces [6–8]. These problems are especially pronounced in low-density areas [9,10]. Issues with detectability are especially acute in invasion fronts such as those for bigheaded carps (*Hypophthalmichys* spp.) in the Mississippi River [9,10].

One way to improve the utility of eDNA as a monitoring tool is to attract fish using sensory attractants, thereby increasing their densities at particular locations to increase detectability. Further, if fish presence is correlated with eDNA copy number, quantification might also be possible. Food has previously been used to attract common carp (*Cyprinus carpio*), rendering them detectable by their eDNA in areas of a lake where they previously were not found because of their low density [11]. However, this technique has not yet been tested on other fishes, including silver carp (*H. molitrix*). In addition to food, another option to attract fish is their pheromones [12,13]; however, this possibility has only been shown

once in a few radio-tagged male common carp and then eDNA was not measured [14]. Notably, all female cyprinids, including invasive common and silver carp, use prostaglandin F2$\alpha$ (PGF2$\alpha$) and its metabolites as a mating pheromone to attract sexually mature male conspecifics [12,15–17]. The species-specificity of the PGF $\alpha$-pheromone appears to be determined by small differences in PGF2$\alpha$ metabolism and body odor composition [14]. The entire "pheromone complex" can be created by implanting carp with PGF2$\alpha$ and then collecting their holding water [14,18,19]. Accurate and sensitive early detection is key to the success of invasive species control programs and enhancing the sensitivity of eDNA-based survey programs is of great interest. In this proof-of-concept study, we tested two questions related to our ability to attract and detect invasive carps:

(1) Can silver carp be attracted to areas where food is being added in such a way that they are more detectable using eDNA and does their presence correlate with eDNA copy number?

(2) Can male common carp be attracted to areas where a mating sex pheromone is being added in such a way that they are more detectable using eDNA and does their presence correlate with the eDNA copy number?

## 2. Materials and Methods

### 2.1. Study Sites and Test Animals

In experiment 1, responses of adult silver carp (unknown sex) to food were monitored in three identical ponds (0.04 ha and 1.3 m deep) at the Southern Illinois University's Touch of Nature Pond Facility (Carbondale, IL, USA). One pond was tested in 2016 and two ponds in 2017. We used new fish each time. Fish were captured using boat-electrofishing (8–12 A and 60 pulses/s) from the Big Muddy River near Murphysboro, IL (USA), transported in aerated river water, implanted with PIT (Passive Integrated Transponder) tags (23 mm HDX PIT tags), and released into study ponds. Stocking of silver carp occurred 1 month prior to experiments in 2016 and 4–7 days prior to experiments in 2017. Twenty silver carp were implanted with PIT tags (23 mm HDX PIT Tags, Oregon RFID, Portland, OR, USA) and stocked in each pond to create an average biomass of 8 kg/ha. To monitor distribution, two PIT tag detection antennas were placed at opposite corners of each pond, each with a detection radius of 6.1 m. Each antenna was connected via a tuning capacitor to a reader (Oregon RFID, Portland, OR, USA), which was powered by a 12 V battery. Ponds were monitored daily, and no mortality was observed. Pond temperatures during the study period averaged 25 °C ± 1.2 SD.

In experiment 2, responses of adult male common carp were monitored in a pond (0.36 ha and 2 m deep) in 2017 at the University of Minnesota's golf course facility (St Paul, MN, USA). Male common carp (N = 15) were boat-electrofished (5–12 A, 120-pulse/s) from Wassermann Lake, Carver County, MN (USA), and their sex was determined by the presence of milt (sperm and seminal fluid). Fish were then transported in aerated lake water, implanted with PIT tags (23 mm HDX PIT tags), and then released in the pond. common carp were stocked 15 days prior to experiments at an average biomass of 6 kg/ha. Two pit tag detection antennas were also placed at opposite corners of this pond to monitor fish distributions. The pond was monitored daily and no mortality was observed.

### 2.2. Experimental Design

After giving the stocked carp 1–3 weeks to acclimate, both experiments commenced with a pre-test period during which time carp distribution was monitored on each side of each pond using PIT tag antennas for two days. During this pre-test period, the side of the pond with the greatest number of carp detections was noted and designated as the control side of the pond. During the test period, the control stimulus (water control) was added to the control side, while the test stimulus (food or pheromone; see below) was added to the other (test) side of the pond. Tests were repeated across two consecutive days.

### 2.3. Experiment 1: Food and Silver Carp

For experiment 1, spirulina, a favored food of silver carp [20], was added to the test side (57 g of powdered spirulina mixed in 2 L of pond water) at the rate of 800 mL/min for 10 min at 20:00 h. The control side of the pond received pond water pumped at the same rate and at the same time. Detections of tagged carp were recorded for both the pre-test and test periods.

### 2.4. Experiment 2: Pheromone and Male Common Carp

For experiment 2, we tested both PGF2$\alpha$ alone and PGF2$\alpha$ pheromone mixture or complex, which contained PGF2$\alpha$ and its metabolites. For the former, we tested a solution of $10^{-9}$ M PGF2$\alpha$ (Cayman Chemical Co., Ann Arbor, MI, USA) [14,18]. For the latter, we created the pheromone complex following established protocols, which entailed implanting carp with PGF2$\alpha$ [18,19,21] and collecting holding water (Supplemental Data S1). The concentration of PGF2$\alpha$ in the mixture was determined to be $10^{-9}$ M by mass spectrometry [11]. Both odors were pumped at a rate of 0.5 mL/min for 40 min and were released on two consecutive days at daybreak (0:600H). The two PGF2$\alpha$ stimuli were evaluated using the same group of male common carp so that these fish experienced 2 days of a pre-test, followed by 2 days of testing with the PGF2$\alpha$ mixture (complex), followed by 1 day of pre-test, and finally 2 days of testing with PGF2$\alpha$ alone.

### 2.5. eDNA Analysis

Five 500 mL water samples were collected from both the test and control sides during pre-test (N = 40) and test (N = 40) periods for both experiments. These samples were collected at the point of stimulus (test side) or water (control side) release in the pond. Test period water samples were collected just after the time period when either the food or pheromone was added (i.e., either 10 min or 40 min). eDNA was then extracted following established protocols by filtering each 500 mL water sample through a glass microfiber filter (47 mm, grade 934-AH, GE Healthcare Life Sciences, Buckinghamshire, UK) using a polyphenylsulfone filter funnel (Pall Corporation, New York, NY, USA) [22]. Filters were folded in half and stored at –80 °C in sterile Whirl–Pak bags (Nasco, Fort Atkinson, WI, USA). Ten filtration controls, consisting of 1 L of distilled water, were also processed with each pond-sampling event (N = 4). Filtration equipment was decontaminated between samples using a 10 min soak in 10% bleach, followed by liberal rinsing with distilled water. Extraction was carried out as previously described by Eichmiller et al. [23]. To measure silver carp DNA in the extracts, we used an established and frequently used qPCR amplification protocol that employs two molecular markers (SC-TM4 and SC-TM5) of silver carp mitochondrial DNA [24]. For common carp, quantitative PCR (qPCR) was also performed as previously described [22] using a qPCR marker specific for common carp that targets mtDNA cytochrome oxidase b gene. Briefly, qPCR reactions (for both silver and common carp) were run in 25μL volumes containing 12.5-μL 2X iTaq Universal Probes Supermix (Bio-Rad, Hercules, CA, USA), bovine serum albumin (New England Biolabs Inc., Ipswich, MA, USA): 10 μg for common carp [23] and 10 mg for silver carp assays [24], 0.5 μM of each primer, 0.375 μM of each probe, and 5 μL of DNA template. Primer and probe sets were run together. Temperature cycling started with an initial denaturation step at 95 °C for 3 min, followed by 40 cycles of 95 °C for 15 s and 60 °C for 1 min. The assay limit of detection (LOD) was defined as the copy number at which 95% of the replicate standard was successfully amplified and was determined to be 30 copies/reaction based on serial dilution studies [23,24]. Duplex qPCR were performed using the StepOnePlus Real-Time PCR System (Life Technologies, Grand Island, NY, USA), and Cq values were automatically determined using the system software. Sample marker concentrations were calculated on a per-run basis. The amplification efficiencies from all plates ranged between 96–106%, which are within normal ranges. No cross contamination was detected in any of the laboratory control samples. The concentration of eDNA was averaged across each sample's replicate reactions prior to data analysis.

### 2.6. Statistical Analyses

To analyze carp distribution, the total number of detections (all individuals combined) on each side of the ponds were first plotted to evaluate overall patterns in distribution (Figure 1; Supplemental Figure S1). We evaluated changes in distribution by examining how relative tendencies (via fold change) to be found on one side compared to the other changed with the addition of stimuli, thereby also meeting assumptions of normality [14]. This approach addressed the lack of normal distribution in raw fish detections. To accomplish this, the relative number of detections on the test side (i.e., the side to which stimuli were added) was calculated as a fold increase in relative distribution for each individual fish. Fold change was calculated by dividing the proportion of an individual's detections on the test side during the test period minus the proportion of an individual's detections on the test side during the pre-test period by the proportion of an individual's detections on the test side during the pre-test period. Initial analysis of experiment 2 (pheromone) showed there to be no difference in the pre-test period distributions of fish prior to the use of the two types of PGF2α pheromone (*t*-test; t = 1.024, df = 1, $p$ = 0.43). Additionally, the response (fold change) of common carp to the two types of PGF2α pheromone was similar (linear mixed-model & likelihood ratio testing; χ2 = 0.013, $p$ = 0.91), so results for both types of pheromone were combined. To compare whether the effects of a stimulus were reduced on second use, a linear mixed-effects model was run with day and stimuli as a fixed effect and individual fish as a random effect, followed by likelihood ratio testing. Across all stimuli, the responses of fish from the first use of a stimulus (day 1) to the second use (day 2) did not change ($\chi^2$ = 1.09, $p$ = 0.30). Therefore day 1 and day 2 of each testing period were combined for further analyses.

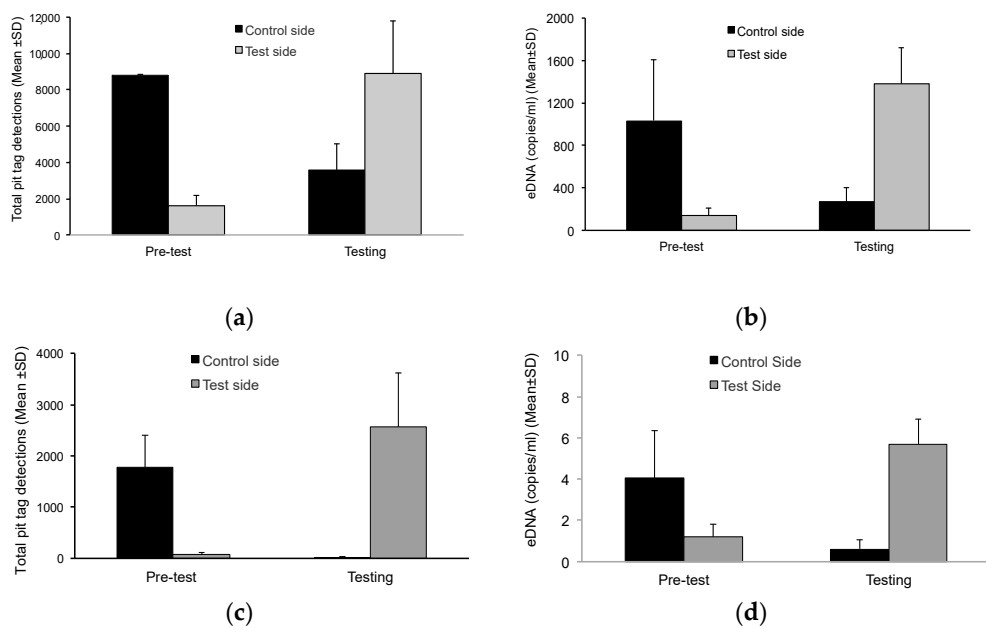

**Figure 1.** (**a**) Total number of detections (Mean ± SD) of silver carp at the test and control sides during pre-test (with no spirulina addition at the test side of the pond) and test periods (during spirulina addition to the test side of the pond) of experiment 1. Values for experiment 1 are all three ponds combined. (**b**) eDNA concentrations (Mean ± SD) at the control and test sides during the pre-test and testing periods of experiment 1. Mean eDNA copy numbers for each side and periods were only available, so the SD value is calculated for the technical replicates of the assay (**c**) Total number of detections (Mean ± SD) of common carp at the test and control sides during pre-test (with no PGF2α addition at test side of the pond) and testing periods (during PGF2α addition to the test side of the pond) experiment 2. (**d**) eDNA concentrations (Mean ± SD) at the control and test sides during the pre-test and testing periods of experiment 2.

To analyze eDNA data, we used Mann–Whitney–Wilcoxon tests to compare eDNA concentrations between control and test sides for both pre-test and testing periods, as these data were not normally distributed. Mann–Whitney–Wilcoxon test could only be conducted for eDNA concentrations during experiment 2 as only mean eDNA copy numbers for each side and periods were available for experiment 1. Fold change was also calculated for eDNA on the test side of the pond as eDNA concentration at the test side during testing minus eDNA concentration at the test side during the pre-test period divided by eDNA concentration at the test side during the pre-test period.

Finally, we examined the relationship between carp density and eDNA concentration for each experiment using Kendall's rank correlations. Mean eDNA copy numbers were non-normal and did not normalize with multiple transformation attempts, likely due to small sample sizes. Therefore, Kendall's rank correlation was used as it does not assume normality. Analyses were performed in R, version 3.2.3.

## 3. Results

### 3.1. Experiment 1: Food and Silver Carp

Silver carp showed a tendency to actively move between both sides of the pond (Supplemental Figure S1a) and this pattern was consistent among individuals. When summarized by total detections on each side of the pond, we found that total detections were higher on the control versus the test side during the pre-test period, with this relationship reversing itself during the test period when the stimulus was added (Figure 1a,b). Mean fold change in the proportion of detections at the test side between pre-test and testing periods was consistently positive for all trials and ponds. The average fold change in carp distribution was a 7.8-fold increase ($\pm$16.9 fold SD). A similar trend was evident with eDNA and average eDNA concentration showed a 4.2-fold increase. There was a significant positive correlation between eDNA copy number and number of carp detections (Figure 2a; $\tau = 1$, $Z = 2.32$, $p = 0.01$).

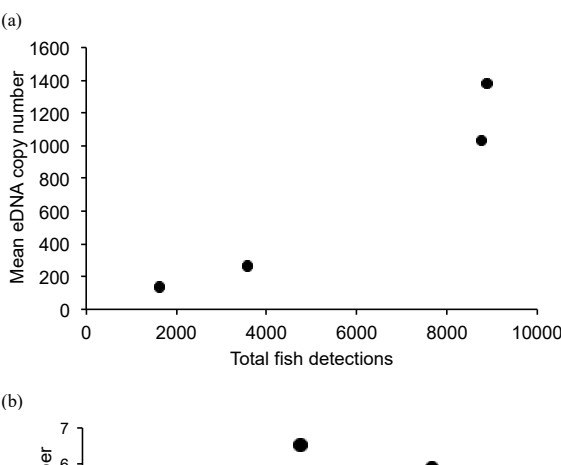

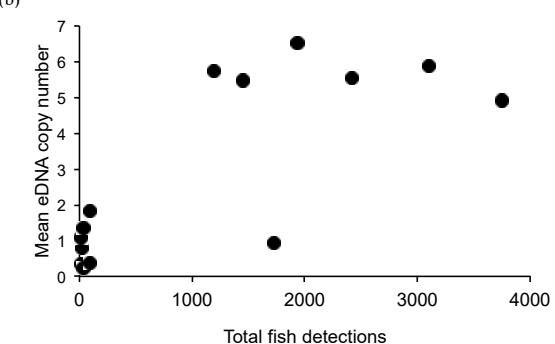

**Figure 2.** Relationship between mean eDNA copy number and the number of fish detections at the test side during each period and day for each attractant type: (**a**) spirulina food for silver carp, (**b**) PGF2$\alpha$ pheromonal sex stimuli for male common carp.

### 3.2. Experiment 2: Sex Pheromone and Male Common Carp

Male common carp showed a tendency to actively move between both sides of the pond each day (Supplemental Figure S1b) and this pattern was consistent among individuals. When summarized by total detections on each side of the pond, we found that total detections were higher on the control versus the test side during the pre-test period, with this relationship reversing itself during the test period when the stimulus was added (Figure 1c,d). Mean fold change in the proportion of detections at the test side between pre-test and testing periods was consistently positive across trials. Fold change averaged a 39.7 ($\pm$33.8 fold SD) increase in test side detections. Fold change in eDNA concentration was positive, averaging an 8.3-fold increase ($\pm$6.8 fold SD). Wilcoxon tests showed significant differences in eDNA concentrations between control and test sides during both pre-test ($W = 7$, $p = 0.04$) and testing ($W = 0$, $p = 0.02$) periods. There was a significant positive relationship between mean eDNA copy number and number of detections (Figure 2b; $\tau = 0.495$, $p = 0.016$).

### 4. Discussion

This proof-of-concept study found that both food and a sex pheromone can be used to induce two species of invasive carp to aggregate in ways that greatly increase eDNA concentrations and presumably make them more detectable. Further, there was a positive association between eDNA copy numbers and carp presence (total fish detections), suggesting eDNA might also be used for the quantification of aggregating carps. Although the PGF2$\alpha$ sex pheromone has yet to be tested with silver carp, it is highly likely that it could also be used on this invasive carp as it has proven to be highly attractive to them in small-scale laboratory tests [14,21]. Of course, large-scale tests of both of these stimuli must now be conducted in flowing river waters. In sum, the possibility of using sensory stimuli to induce wild fishes to aggregate so that they can be easily and accurately measured appears promising for possible use in invasive carp management.

The possibility of using food to attract a fish species so it can be measured more easily and accurately has several distinct advantages that might make it practical. In particular, food can be inexpensive and easy to add to water and is likely to attract both sexes and multiple developmental stages (e.g., juvenile vs. adult). However, effectiveness is likely to be less if fish are not hungry because the waters are cold or natural food is abundant [11]. However, as previously shown for common carp, the utility of food as an attractant can be enhanced if it is predictably added to a specific location for an extended period of time to condition (train) fish to come to that location; indeed, the attraction was seen to increase dramatically after a week of continuously baiting [11]. Ghosal et al. [11] speculated that the act of food consumption stimulated eDNA shedding rates and could increase detectability. This was evident in our study, too, that the eDNA concentrations were higher on the stimulus side during food baiting when compared to pheromone baiting. This could be due to higher behavioral activity (buccal pumping, jostling/pushing) of fish while feeding and might also be associated with higher excretion rates. Similar patterns showing a direct positive correlation between behavioral activity and eDNA concentrations were observed by Thallinger et al. [8] for several freshwater species, including four species of Salmonids, two Cyprinids, and one Sculpin. Further exploration of food attractants in invasive carp holds promise for improving the effectiveness of eDNA monitoring and fish removal programs, and different types of food and conditioning regimes should be explored.

Sex pheromones could complement the use of food as a carp attractant for eDNA monitoring across the seasons because pheromones may, as indicated by our proof-of-concept study, be a highly effective attractant during the spawning season. Notably, fish often reduce their feeding activity when reproductively active [25], although it is not clear if this is true for the carps. However, the success rate of inducing aggregations may also vary during the mating season. Johnson et al. [26] suggested that traps baited with sex pheromone component of sea lamprey had a higher capture rate in streams with a lower number of spawning events because of fewer competing natural pheromone sources. The

fact that sex pheromones are likely to attract only mature conspecific males can be both a strength as well as a weakness. Notably, in our study, there was also a strong relationship between detections of male common carp and eDNA copy numbers which appear to allow for the possibility of quantifying fish density if the active space of the pheromone were calculated. Tests are now required in flowing waters to evaluate the ability of sex pheromones to attract invasive carp.

The promising results of the present proof-of-concept study strongly suggest that future studies should explore how sensory stimuli, including other types of food and pheromones, might be used to attract fish so that their presence and relative abundance might be accurately and precisely measured by eDNA. Such an approach could remove some of the uncertainties associated with determining the best places and times to sample with both eDNA and traditional fisheries gears. To date, an integrative approach using a stimulus and eDNA to induce and detect aggregations of fish has only been tested in lentic systems [11], (the present study); it now requires evaluation in lotic systems. The role of species-specific responses as well as environmental factors including water turbulence, temperature, and current on the relationship between fish distribution, density, and eDNA concentrations should also be investigated to increase detection sensitivity further. For food, it would be insightful to include long-term conditioning as it has been used successfully in lakes [11]. Conversely, habituation could be confounding for pheromones. It is especially important that future work evaluate this concept in large, natural lotic systems where additional variables would be expected to influence the ability to attract desired species. Possible effects of water current and chemistry, as well as the presence of non-target fish species, need to be investigated in particular.

## 5. Conclusions

Coupling the addition of sensory attractants such as food and sex pheromones with eDNA measurements can enhance both the utility and sensitivity of eDNA surveillance techniques for invasive carp, especially in low-density areas where carp densities and eDNA levels may be very low. The same technique could also be used to enhance removal efforts by concentrating fish at target locations.

**Supplementary Materials:** The following supporting information can be downloaded at: https://www. mdpi.com/article/10.3390/fishes7040176/s1, Supplementary Data S1: Protocol for preparing PGF2$\alpha$ odor. Figure S1a Total number of detections (Mean $\pm$ SD) of silver carp by day at the test and control sides during pre-test (with no spirulina addition at the test side of the pond) and testing periods (during spirulina addition at the test side of the pond) pooled across ponds (N = 3). Figure S1b Total number of detections (Mean) of common carp at the test and control sides during pre-test (with no PGF2$\alpha$ addition at test side of the pond) and testing periods (during PGF2$\alpha$ addition at test side of the pond).

**Author Contributions:** Conceptualization, R.G., A.A.C. and P.W.S.; methodology, R.G. and A.A.C.; formal analysis, R.G. and A.A.C.; resources, P.W.S.; writing—original draft preparation, R.G., A.A.C. and P.W.S.; writing—review and editing, R.G., A.A.C. and P.W.S.; visualization, R.G., A.A.C. and P.W.S.; funding acquisition, PWS. All authors have read and agreed to the published version of the manuscript.

**Funding:** This research was funded by the Minnesota Environment and Natural Resources Trust Fund [Sponsored Project: 00033145—Aquatic Invasive Species (AIS)] as recommended by the Legislative-Citizen Commission on Minnesota Resources.

**Institutional Review Board Statement:** The study was approved by the Institutional Animal Care and Use Committee of the University of Minnesota protocol#1009A88894.

**Data Availability Statement:** All the data is available within the manuscript and the Supplementary Materials.

**Acknowledgments:** The authors would like to thank Kasey Yallaly and Justin Seibert for their assistance collecting silver carp and Reid Swanson and Justine Dauphinais for their help in electrofishing the common carp. The authors would like to thank Jessica Eichmiller and Grace Van Susteren for their assistance during fieldwork and Ping Wang for his help conducting eDNA analyses.

**Conflicts of Interest:** The authors declare no conflict of interest. The funders had no role in the design of the study, in the collection, analyses, or interpretation of data, in the writing of the manuscript, or in the decision to publish the results.

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
