# Peer review of "Proof-of-Concept Studies Demonstrate That Food and Pheromone Stimuli Can Be Used to Attract Invasive Carp So Their Presence Can Be Readily Measured Using Environmental DNA"

_fishes, doi:10.3390/fishes7040176_

Round 1
Reviewer 1 Report
Ghosal et al. conducted what they referred to as a “pilot study” to test whether food stimulus can be used to concentrate silver carp and sex pheromone can be used to concentrate common carp. This aspect of the study is not particularly novel, and the responses of either of these fish species to the respective stimuli is not surprising. The novelty of this pilot study is that the authors used eDNA PCR assays to demonstrate that “detectability” of these fish increased as a result of concentrating the fish using these stimuli. The most surprising thing that I found in this study is that, in relatively small holding ponds the researchers were able to record differences in eDNA concentrations of the magnitude they reported. In point of fact, the authors did detect significant differences in eDNA concentrations throughout the ponds. Though the magnitude of the differences in concentrations on opposite sides of the ponds seemed to be on the order of two to eight-fold, judging from figure 1b and 1d. Therefore, it seems that the fish were “detectable” on both sides (control and test sides) of the ponds, both pre-test and post-test. The authors correctly point out that these results need to be repeated in a flowing river system. They do not point out that their hypothesis, that these stimuli increase detectability of target fish species, needs to also be tested in an open natural environment system where fish are not limited to very small and artificial pond habitats. The number of replicates of each of the studies is rather small, though the results are clear. I’m really struggling to decide if the novelty and interest level of this study warrants its publication. As a short note, maybe. As a full length research article, I think probably not.
I found figures 1 and 3 to be useful and informative as I read the manuscript. I don’t think figures 2 and 4 are warranted for their information content, which I think could just as easily be reported in the text. I also think that the percent changes in detections in the text should be changed to fold increases in detection. So for example, in line 203, a 418% increase should be reported as a 4.2 fold increase. In line 221, an 831% increase should be reported as an 8.3 fold increase. I’d like to see the authors expand their discussion to include the limitations of conducting detections studies on very large fish in very small holding ponds and the implications for moving a study like this to very large natural riverine habitats where the authors are proposing that these methods will have practical applications. One important point that the authors make is that a small number of captive fish become quickly conditioned to stimuli, like a food source. The authors don’t discuss if this behavioral conditioning would be the same in a larger open riverine system where fish would potentially range much further. What impact could that have on the detectability of fish with versus without attractant? Does the literature have anything to offer on that question?
The methods are very concise and omit a lot of detail, relying on references to published protocols instead. I’d like to see some more details on the molecular protocols. For example, what instrument and reagents were used, and I think also the PCR conditions. Were both silver and common carp assays run on a quantitative PCR machine? That is not clearly stated in lines 134-136.
Author Response
Dear reviewers,
Thanks for considering our manuscript on “Proof-of-concept studies demonstrate that food and pheromone stimuli can be used to attract invasive carp so their presence can be readily measured using environmental DNA”. We greatly appreciate the feedback from all the three reviewers and have addressed the concerns in the revised version of the manuscript. In this document, we have provided specific responses to each of the concerns raised by all the three reviewers.
Reviewer#1
Ghosal et al. conducted what they referred to as a “pilot study” to test whether food stimulus can be used to concentrate silver carp and sex pheromone can be used to concentrate common carp. This aspect of the study is not particularly novel, and the responses of either of these fish species to the respective stimuli is not surprising. The novelty of this pilot study is that the authors used eDNA PCR assays to demonstrate that “detectability” of these fish increased as a result of concentrating the fish using these stimuli. The most surprising thing that I found in this study is that, in relatively small holding ponds the researchers were able to record differences in eDNA concentrations of the magnitude they reported. In point of fact, the authors did detect significant differences in eDNA concentrations throughout the ponds. Though the magnitude of the differences in concentrations on opposite sides of the ponds seemed to be on the order of two to eight-fold, judging from figure 1b and 1d. Therefore, it seems that the fish were “detectable” on both sides (control and test sides) of the ponds, both pre-test and post-test. The authors correctly point out that these results need to be repeated in a flowing river system. They do not point out that their hypothesis, that these stimuli increase detectability of target fish species, needs to also be tested in an open natural environment system where fish are not limited to very small and artificial pond habitats. The number of replicates of each of the studies is rather small, though the results are clear. I’m really struggling to decide if the novelty and interest level of this study warrants its publication. As a short note, maybe. As a full length research article, I think probably not.
Response: Thank you for these comments. We have clarified in the manuscript that we are not testing for open systems. We have also added sentences to the discussion to point out the need to this study to be repeated with additional replicates and in open systems. (lines 367-374).
I found figures 1 and 3 to be useful and informative as I read the manuscript. I don’t think figures 2 and 4 are warranted for their information content, which I think could just as easily be reported in the text. I also think that the percent changes in detections in the text should be changed to fold increases in detection. So for example, in line 203, a 418% increase should be reported as a 4.2 fold increase. In line 221, an 831% increase should be reported as an 8.3 fold increase.
Response: We have removed figures 2 and 4 and now present the values from these within the text. We have also changed our verbiage from percent change to fold-increase (lines 191-195; 213-217).
I’d like to see the authors expand their discussion to include the limitations of conducting detections studies on very large fish in very small holding ponds and the implications for moving a study like this to very large natural riverine habitats where the authors are proposing that these methods will have practical applications. One important point that the authors make is that a small number of captive fish become quickly conditioned to stimuli, like a food source. The authors don’t discuss if this behavioral conditioning would be the same in a larger open riverine system where fish would potentially range much further. What impact could that have on the detectability of fish with versus without attractant? Does the literature have anything to offer on that question?
Response: We agree with the reviewer that there may be several limitations in conducting a similar study on very large fish or in a riverine habitat. However, similar work using wild or free-ranging fish of larger size (100-150 kg) has been conducted on Common Carp (Ghosal et al. 2018), and this study was able to induce large aggregations using food in a natural lake (area of 67 ha). Conditioning can extend over days for open water, larger systems. For example, Ghosal et al. (2018) showed that fish aggregated near the food stimulus only after 96 hours (4 days) of continuous baiting. Whereas in closed, smaller systems, aggregation may happen faster. However, the riverine habitats will be a challenge, and to the best of our knowledge, no studies have investigated the impact of baiting on fish behavior in a large river system. We have added some caveats to the paper at lines 367-374.
The methods are very concise and omit a lot of detail, relying on references to published protocols instead. I’d like to see some more details on the molecular protocols. For example, what instrument and reagents were used, and I think also the PCR conditions. Were both silver and common carp assays run on a quantitative PCR machine? That is not clearly stated in lines 134-136.
Response: We have now included specific details on the molecular protocols. Please refer to the revised version of the manuscript. We mention that both Silver and Common carp assays were done on a quantitative PCR machine and have clarified that in the revised version of the manuscript (lines 159-178).
Reviewer 2 Report
The manuscript is well presented and written clearly. I foud the topic of the research interesting, which adds valuable information to the current knownledge. A few minor adjustements required:
Font size needs to be adjusted from line 61 to line 92.
In my version, the impagination of the figures need to be adjusted (also in supplementary material).
line197: remove "that"
line216: remove "that"
Author Response
Dear reviewers,
Thanks for considering our manuscript on “Proof-of-concept studies demonstrate that food and pheromone stimuli can be used to attract invasive carp so their presence can be readily measured using environmental DNA”. We greatly appreciate the feedback from all the three reviewers and have addressed the concerns in the revised version of the manuscript. In this document, we have provided specific responses to each of the concerns raised by all the three reviewers.
Reviewer#2
The manuscript is well presented and written clearly. I foud the topic of the research interesting, which adds valuable information to the current knownledge. A few minor adjustements required:
Response: We thank the reviewer for appreciating our work.
Font size needs to be adjusted from line 61 to line 92.
Response: We have now adjusted the font size in the revised version of the manuscript.
In my version, the impagination of the figures need to be adjusted (also in supplementary material).
Response: As suggested by Reviewer#1, we have now removed the figures 2 and 4, and the placement and numbering of the figures have been adjusted accordingly.
Line 197: remove "that"
Response: Thanks; we have now removed the word in revised version of the manuscript.
line 216: remove "that"
Response: Thanks; we have now removed the word in revised version of the manuscript.
Reviewer 3 Report
The manuscript analyzed invasive carp by eDNA by testing whether food or pheromones might be used to attract invasive carp. There are several indications to improve as follows:
The authors should provide in detail the condition for PCR amplification in lines 134-136. In addition, authors should show a range of food and pheromone concentrations examined and expose time of food and pheromone.
As the authors pointed, large-scale tests of both of these stimuli must be executed. Attracting carp using food or pheromones may be challenging in wild environments due to the ecological relationships with other species and other physicochemical environmental factors.
The following is minor point should be revised:
Line 14: Please write the full name of PIT
Line 25: Change “Bigheaded carp” to “Silver Carp”
Line 197: The word ‘that’ was written twice.
Author Response
Dear reviewers,
Thanks for considering our manuscript on “Proof-of-concept studies demonstrate that food and pheromone stimuli can be used to attract invasive carp so their presence can be readily measured using environmental DNA”. We greatly appreciate the feedback from all the three reviewers and have addressed the concerns in the revised version of the manuscript. In this document, we have provided specific responses to each of the concerns raised by all the three reviewers.
Reviewer#3
The manuscript analyzed invasive carp by eDNA by testing whether food or pheromones might be used to attract invasive carp. There are several indications to improve as follows:
The authors should provide in detail the condition for PCR amplification in lines 134-136.
Response: We have now included specific details on the molecular protocols. Please refer to the revised version of the manuscript (lines 159-178)
In addition, authors should show a range of food and pheromone concentrations examined and expose time of food and pheromone.
Response: We tested only one concentration for the food stimulus (57 g of powdered Spirulina mixed in 2 L of pond water) for the Silver Carp. Higher concentrations were clogging the tubes of the pump, and thus, were avoided. For pheromone release, we used only one concentration of PGF2α (10-9M), as this had been proven by many earlier studies (Lim and Sorensen, 2010, 2011, 2012) to be potent
As the authors pointed, large-scale tests of both of these stimuli must be executed. Attracting carp using food or pheromones may be challenging in wild environments due to the ecological relationships with other species and other physicochemical environmental factors.
Response: We agree with the reviewer that there may be several limitations in conducting a similar study in a wild environment. We have also added sentences to the discussion to point out the need to this study to be repeated in open systems (lines 367-374).
The following is minor point should be revised:
Line 14: Please write the full name of PIT
Response: Thanks; we have added the full form of PIT (line 77) in revised version of the manuscript.
Line 25: Change “Bigheaded carp” to “Silver Carp”
Response: Thanks; we have changed to “Silver Carp” in revised version of the manuscript.
Line 197: The word ‘that’ was written twice.
Response: Thanks; we have now removed the word “that” in revised version of the manuscript